# A Colloidal-Quantum-Dot Integrated U-Shape Micro-Light-Emitting-Diode and Its Photonic Characteristics

**DOI:** 10.3390/nano14110938

**Published:** 2024-05-27

**Authors:** Yu-Ming Jao, Bo-Ming Huang, Ching Chang, Fang-Zhong Lin, Guan-Ying Lee, Chung-Ping Huang, Hao-Chung Kuo, Min-Hsiung Shih, Chien-Chung Lin

**Affiliations:** 1Graduate Institute of Photonics and Optoelectronics, National Taiwan University, Taipei 10617, Taiwan; r10941132@ntu.edu.tw (Y.-M.J.); r10941135@ntu.edu.tw (B.-M.H.); r10941136@ntu.edu.tw (C.C.); 2Institute of Lighting and Energy Photonics, College of Photonics, National Yang Ming Chiao Tung University, Tainan City 71150, Taiwan; lfz199712.cop09g@nctu.edu.tw (F.-Z.L.); nctymu309806010.pt09@nycu.edu.tw (G.-Y.L.); 3Graduate Program of College of Photonics, College of Photonics, National Yang Ming Chiao Tung University, Tainan City 71150, Taiwan; ping.pt05@nycu.edu.tw; 4Department of Photonics, National Yang Ming Chiao Tung University, Hsinchu 30010, Taiwan; 5Semiconductor Research Center, Hon-Hai Research Institute, Taipei 114699, Taiwan; 6Research Center for Applied Sciences, Academia Sinica, Taipei 115201, Taiwan; mhshih@gate.sinica.edu.tw; 7Department of Electrical Engineering, National Taiwan University, Taipei 10617, Taiwan

**Keywords:** micro LEDs, colloidal quantum dots, color conversion efficiency

## Abstract

A special micro LED whose light emitting area is laid out in a U-like shape is fabricated and integrated with colloidal quantum dots (CQDs). An inkjet-type machine directly dispenses the CQD layer to the central courtyard-like area of this U-shape micro LED. The blue photons emitted by the U-shape mesa with InGaN/GaN quantum wells can excite the CQDs at the central courtyard area and be converted into green or red ones. The U-shape micro LEDs are coated with Al_2_O_3_ by an atomic layer deposition system and exhibit moderate external quantum efficiency (6.51% max.) and high surface recombination because of their long peripheries. Low-temperature measurement also confirms the recovery of the external quantum efficiency due to lower non-radiative recombination from the exposed surfaces. The color conversion efficiency brought by the CQD layer can be as high as 33.90%. A further continuous CQD aging test, which was evaluated by the strength of the CQD emission, under current densities of 100 A/cm^2^ and 200 A/cm^2^ injected into the micro LED, showed a lifetime extension of the unprotected CQD emission up to 1321 min in the U-shape device compared to a 39 min lifetime in the traditional case, where the same CQD layer was placed on the top surface of a squared LED.

## 1. Introduction

In recent years, the rise of the Internet has brought tremendous changes to our daily life. The wide spread of mobile devices that can be connected via the Internet leads to a high demand for a good machine–human interface so that we can obtain information from the Internet more easily. This interface should be interactive, fast responsive, and usually with great graphic effects. To meet these requests, a high-quality, full-color display is needed. In the past few years, research teams around the world have pursued this goal very actively. Many technologies, like liquid crystal display (LCD) and organic light emitting diodes (OLEDs), have been introduced to fulfill such needs [1,2,3]. But many believe that we need a more efficient, power-saving, and self-emissive technology to address the everlasting demand on this machine–human interface. The currently available displays can accomplish part of these stringent requirements, but there are always some inadequacies that can lead to unsuccessful products. Semiconductor-based micro LEDs to fill this gap have been mentioned recently [4]. In this technology, each pixel of a display is a self-emissive semiconductor unit and the basic primary colors (red, green, and blue) should be presented in the same pixel. Highly efficient and scalable semiconductor devices have been the main driving forces for our electronic industry. Thus, it is highly anticipated that this success can be transferred to the photonic world; but to achieve this, many innovations are needed.

There are two major roads to carry out this semiconductor-based technology at present. One is to use individual semiconductor micro LEDs and assemble them into arrays of pixels [5,6,7]. In this method, different active materials (like InGaN and AlGaInP) are needed to generate different colors. The other method is to use a monochromatic micro LED array to pump and excite a layer of patterned color converting material to form a full-color display [8]. Both methods have attracted considerable attention and they have their own pros and cons. The necessity to have different active materials plays a critical role for the first method. Although the QW’s width can modify the emission color, the primary material (the composition or species) must be significantly adjusted to show a dramatic color tuning. For example, between blue (hν = 2.755 eV) and red (hν = 1.968 eV) photons, there is a 0.787 eV difference in emission photon energy, which is very difficult to be covered by any single material as a light-emitting layer. Therefore, if we want to achieve full-color pixels in one display solely by semiconductor LED devices, it is easier to use different active materials such as InGaN and AlGaInP, manufacture the blue, green, or red color micro LEDs from different epitaxial wafers, and then bond them separately onto the same substrate [4,5]. Many companies have adapted this method to produce the first generation of micro LED TVs, but when the pixel size reduces and the number of pixels increases, this method becomes impractical and technically almost impossible to implement due to the lack of suitable transfer technology. The high-yield assembly and transfer of such a small-sized micro LED is a non-trivial task because the traditional pick-and-place method relies on a large, flat-top surface of the device for a firm grip. Another area for improvement is the reduction of the device’s external quantum efficiency (EQE) when the device’s size is reduced to below 5 microns [9,10,11]. This reduction in EQE is mainly due to the increased sidewall surface-to-volume ratio, significantly increasing non-radiative recombination along the sidewalls for the electron/hole pairs [12]. Meanwhile, the color conversion efficiency (CCE) and the lifetime issues of the conversion materials are mostly discussed for the color conversion method. The color conversion material has to be optically efficient for converting the high-energy photons (like blue ones) into green or red photons and be formatted into small pixels simultaneously. Since many of these materials, such as inorganic quantum dots, fluorescent dyes, or phosphors, are not photo-lithographically sensitive, extra efforts are needed to incorporate these active materials with certain kinds of UV-sensitive resins [8,13]. The small pixel requirement in the future microdisplay is also a severe problem for the color conversion method because the pixel size of the color conversion layer (CCL) can be defined either by photolithography or direct deposit of the color conversion material. It takes a lot of effort and chemistry knowledge to provide fine-pitch photolithography in CCL [14], and a technological innovation called the electrohydrodynamic (EHD) mechanism to shrink the color conversion material to sub-micron pixels in the direct deposition method [15]. In the past, most conversion materials were placed directly on the top of the pumping micro LEDs, and this arrangement could be optically ineffective when there is no other optical component (such as a reflecting mirror) in the device, or when the conversion material is optically thin [16,17,18]. Another problem with traditional CCL deposition is the direct contact with or close to the micro LEDs. The pumping micro LEDs represent the biggest heat source in the display module, and overheating can quickly degrade the illumination efficiency of the CCL. To tackle these problems, we design a novel U-shape micro LED and integrate it with colloidal quantum dots (CQDs) as the color conversion material in this study. The unique shape of the micro LED and the special location of the color conversion layer can greatly solve the problems that we just mentioned. This U-shape structure of micro LED is intended to accommodate any kind of nanoparticle (such as perovskite QDs, inorganic QDs, etc.), which can convert the blue/UV photons into longer wavelength ones. As long as the deposition can be performed in the central courtyard with some precision, the deposited color conversion material will be able to receive and convert the blue or UV photons properly. In the following content, the details of device fabrication and photonic characteristics will be demonstrated and the superiority of the design will be explored.

## 2. Device Fabrication and Measurement

This section focuses on the U-shape micro LED fabrication. The wafer was purchased from the external vendor (Epileds Technologies Inc., Tainan, Taiwan) and was grown by a metal organo-chemical vapor deposition system. The quantum well of the U-shape LED, which provides the blue photons in this study, is composed of 25% of indium in InGaN, with the quantum well width around 4 nm and quantum barrier around 9 nm. The colloidal quantum dots were purchased from an external vendor (Unique Materials Co., Ltd., Taichung, Taiwan). The dots were composed of a CdSe/ZnS core-shell structure. The peak emission wavelength is around 525 nm and the FWHM is around 21 nm. A layer of indium–tin-oxide (ITO) was deposited on the epitaxial wafer. The wafer was then patterned by a photolithography step to have a U-shape metal deposited and lifted off. The metal layer, which is used as the p-type contact, is composed of Ni (20 nm) and Au (20 nm). A rapid thermal annealing (RTA) process was performed at 500 °C for 10 min and an oxygen-containing environment was needed to form a good p-type contact [19]. The mesa etch has two steps: first is a HCl (37%) wet etch to remove the uncovered ITO region and the second is an inductively coupled plasma (ICP) dry etch to remove the unpatterned InGaN and GaN material. The etch depth to reveal the n-type GaN surface is 2 μm. The ICP etch gases are Ar (5 sccm), BCl_3_ (30 sccm), and Cl_2_ (20 sccm), with an RF power of 150 watts. A Cr (30 nm)/Au (300 nm) layer was then evaporated by a vacuum evaporation system and also patterned by a photolithography + lift-off process. The metal evaporation chamber is kept at 2.25 × 10^−5^ Torr and the deposition rate for Cr is around 0.1 nm/s and for Au, it is below 0.6 nm/s. The samples were attached to a rotating plate to ensure thickness uniformity in deposition. Two different dielectric layers were applied for the exposed mesa sidewall protection and electrical isolation. An atomic layer deposition (ALD) system was used first for the deposition of a 30 nm Al_2_O_3_ layer and then followed by a 400 nm PECVD SiO_2_ deposition. We used a Picosun^®^ R-200 Advanced ALD system made by Picosun Oy, Finland, to deposit the Al_2_O_3_ passivation layer. Two different chemicals, Al(CH_3_)_3_ (trimethylaluminum, TMA) and H_2_O, are used as precursors and the deposition temperature is 300 °C. The SiO_2_ layer was deposited in a PECVD system. The deposition temperature is 300 °C and we use SiH_4_ and N_2_O as the reactive gases. The chamber is pumped down to 1000 mTorr and the RF power is 50 watts. The deposition rate of the SiO_2_ is around 60 nm per minute. The metal contact open can be made by a buffered oxide etch and DI water (1:6) mixture. A final Al (50 nm)/Au (500 nm) metal deposition is needed to properly cover the outer periphery and top surface of the U-shape device. This metal layer will redirect the blue photons back to the central area. The color conversion material, i.e., the CQD layer in this study, can be directly cast onto the central “courtyard” area by the Aerosol Jet spraying system [20]. The Aerosol Jet system was manufactured by Optomec, Albuquerque, NM, USA (Aerosol Jet^®^ 300), and the computer controlled process can provide one or multiple depositions of CQDs on the same spot [16]. Figure 1a illustrates the schematic diagram of our finished device with a CQD layer deposited in the central courtyard area. Figure 1b,c are the finished devices under the optical microscope and the fluorescent optical microscope (FLOM). The unique arrangement of the CQD layer can alleviate the CQD layer from high heat generated by the InGaN/GaN device. In the past, the CQD layer was placed directly on the flat top of the square or circular LED. However, we found that this arrangement style can lead to a direct heat transfer from the semiconductor device toward the CQD layer and eventually kill the CQD layer’s illumination. The central courtyard-like area of a U-shape diode can provide a suitable space for storing the CQD layer. The mesa surrounding the CQD layer can provide blue photons to excite the CQD. At the same time, the heat generated by the InGaN/GaN U-shape diode can be dissipated without passing through the CQD layer, and thus the CQD’s lifetime can be extended. The shape of the device mesa was determined to use a U-shape due to the concern of the deposition of CQD particles. The CQD solution cannot permeate into an encircled space if the host solution has a strong surface tension. Therefore, we believe a partially open U-shape shall be helpful to alleviate this situation.

## 3. Experimental Results

### 3.1. LIV (Luminance–Current–Voltage) Measurements and Spectra of U-Shape Micro LEDs

To measure these U-shape micro LEDs, we need to categorize them first. Because the central area that is surrounded by three branches of the mesa is designed to be a square, we can identify the device by the side length of this courtyard-like area and its surrounding mesa width. This central square courtyard area surrounded by the U-shape mesa can host CQD material nicely and the metal layout covering the mesa top and outer rim regions can reflect the blue photons to the central courtyard area to better excite the CQDs in the integrated device. The larger this courtyard area is, the larger the actual illuminating active area will be. The calculation of the total mesa area and the periphery in a specific U-shape micro LED can be shown as the periphery length = 6 × (*a* + *w*) and the area of mesa = *w* × (3*a* + 2*w*), where the mesa width is w and the side length of the courtyard area is a; the periphery-to-mesa-surface ratio (called the P/S ratio [12]) can be calculated as:(1)$$P/S\_ratio=\frac{6\left(a+w\right)}{w\left(3a+2w\right)}\;$$

In this study, the width of the device mesa is between 10 and 15 μm. Several courtyard sizes are designed to explore the device characteristics: 40, 50, 60, and 70 μm, respectively.

After the U-shape devices are fabricated, their spectra can be measured under different current levels. The devices can be probed on a wafer with a proper electrical and photo-detection setup, as we mentioned in Section 2. Figure 2a shows the light output and diode voltage versus injection current density of 10-micron mesa U-shape devices with different courtyard sizes. The larger the courtyard is, the higher the expected output power due to the larger light emission area. We would like to point out that the electrical isolation between the CQD layer and the U-shape diode was achieved by the dielectric layer covering the top surface of the semiconductor. This dielectric layer is the same passivation layer that we deposited during the ALD and PECVD steps and the total thickness is 430 nm. A further measurement of the devices before and after the CQD dispense shows that the diode series resistances are unaffected, which means good isolation between the CQD layer and the U-shape diode. This electrical isolation also ensures that our CQD layer does not receive any electrical injection and only receives the photonic excitation from the U-shape micro LED’s blue photons. The current-dependent spectra were also taken, as shown in Figure 2b. In Figure 2b, a blue shift in the peak wavelength can be observed and the difference (∆λ) of emission peaks can be as large as 5.5 nm between 1 mA and 10 mA. This phenomenon can be explained by the screening effect and the reduction of the inherent quantum-confined Stark effect (QCSE) or enhanced band-filling effects [21,22,23,24,25,26].

### 3.2. EQE of a U-Shape Micro LED

To test the light illumination capability of the individual micro LED, we devised a method by using a homemade probe that consists of a GaAs single junction solar cell chip and a metal probe attached to a linear stage with XYZ translation. The solar cell chip detects emitted photons from the micro LED under measurement. The detected photons can then be converted to photocurrents and measured by an external source meter. The details were first described in [9], and the corresponding EQE can be obtained from the photocurrent readings of this solar cell chip. The small volume of the solar cell chip ensures its proximity to the micro LED, and this proximity can enhance the collection of the emitted photons from the micro LED. The measured EQE was obtained from the photocurrent detected by using this formula to calculate the EQE of the U-shape micro LED [9]:(2)EQE=Photons EmittedInjected Electron−hole pairs=PopthνIdiode/q=qhνPoptIdiode=IphIdiode×ηsolar  
where *I_diode_* is the electrical current injected into the micro LED, *P_opt_* is the optical power emitted by the U-shape device, *I_ph_* is the measured photocurrent from the solar cell, and *η_solar_* is the measured quantum efficiency of the solar cell. The EQE can then be numerically evaluated by the *ABC* model, which uses three mechanisms to describe the major carrier photon interactions in a LED: a non-radiative recombination of the Shockley–Read–Hall (SRH) process, a radiative bimolecular recombination by an electron-hole pair, and a non-radiative recombination due to the Auger process [27]. In the ideal situation, these three mechanisms are sufficient to describe the quantum efficiency change due to current injection. However, if there is an extra leakage source happening in the device, such as the carrier overflow, we need to modify the original *ABC* model to better fit the experimental results [9]. The overall formula then becomes [9]:(3)$$EQE=\frac{\eta_{LEE}\left(1-\beta n\right)Bn^{2}}{\left(An+Bn^{2}+Cn^{3}\right)}$$
where *A*, *B*, and *C* are the SRH coefficient, the bimolecular coefficient for radiative recombination, and the Auger recombination coefficient, respectively, and *β* is the current leakage parameter [9]. *η_LEE_* is the light extraction efficiency and can be treated as a constant during calculation [28,29]. These parameters can be numerically fitted into the measured data points. Thus, Equation (3) is derived from the basic definition of the leakage current during the carrier injection [30,31,32,33]:(4)$$J_{leak}=\beta \delta nJ_{total}=\beta nJ_{total}$$
where *δn* ≈ *n* is the excess carrier concentration that is injected into the device and *β* is related to the carrier mobility coefficients and the part of the carrier population that can spill over the active region [33]. This leakage term needs to be included in the original *ABC* model:(5)$$J_{total}=J_{ABC}+J_{leak}=qt\left(An+Bn^{2}+Cn^{3}\right)+\beta nJ_{total}$$
(6)$$J_{total}=\frac{qt\left(An+Bn^{2}+Cn^{3}\right)}{\left(1-\beta n\right)}\;$$

Because the total current is increased substantially by this leakage (through 1 − *β*), the EQE needs to be adjusted accordingly, as shown in Equation (3).

In Figure 3a–d, the measured EQE was plotted against the current density in the U-shape micro LEDs. The devices were measured at room temperature (300 K), and they are among the best devices that were fabricated in this study. The peak EQE of a 40 μm-courtyard device is 2.04% (at 143 A/cm^2^) and that of a 50 μm-courtyard device is 6.51% (at 32.35 A/cm^2^). Meanwhile, a 60 μm-courtyard device has an EQE_max_ of 6.15% at 25 A/cm^2^ and a 70 μm-courtyard device’s EQE_max_ is 6.26% at 30.43 A/cm^2^. Figure 3e shows the normalized EQE results for the best devices among various sizes of the courtyard area. On average, the smaller devices had worse EQE_max_ values than the larger ones. The differences could be as large as 25%, and we believe this difference arises from the high P/S ratio of the smaller devices. The higher P/S ratio can contribute to higher non-radiative recombination of the sidewall and influence the eventual EQE through the SRH recombination coefficient in the ABC model [9,10,12,34].

### 3.3. Photonic Characterization with CQDs

After CQDs are dispensed into the courtyard area, the device becomes a color-converted micro LED. The optical spectrum of the device changes from a monochromatic peak to a dual peaked one (blue and CQD emission). If we tracked both blue and green peaks together, a plot under different injection currents can be obtained in Figure 4a. Two U-shape devices with 10 μm mesas and 60 and 70 μm courtyard sizes were tested. From the measured result, the CQD emission varies very little across different current levels, while the pumping U-shape devices change their wavelengths more than 5 nm. From the spectra of Figure 4a, the emission peaks of the U-shape device and the CQD layer can be traced and plotted in Figure 4b. The blue peaks moved towards a shorter wavelength under increased current injection, as we expected and observed prior to the CQD dispense. In the two devices that we examined, the blue shifts were 6.64 nm and 5.53 nm for 60 μm and 70 μm devices, respectively. Under the same current range (from 1 mA to 10 mA), the CQD emission peaks are relatively stable and move less than 0.5 nm (−0.21 nm for the 60 μm device and 0.08 nm for the 70 μm one). This phenomenon exemplifies one of the biggest advantages of the CQD: the color stability under various excitation conditions.

The CCE can be obtained by comparison of the two spectra of the same device with and without CQDs. As shown in Figure 5a, there are two spectra taken from the same device. One is measured before the CQD dispense and is named $I_{ref}(\lambda )$ in the plot. The other is measured after the CQD dispense and is named $I_{QD}(\lambda )$. The area underneath the spectrum represents the number of photons at the wavelength range. To obtain the color conversion efficiency of the CQD layer, we can write down the formula as:(7)$$CCE=\frac{\#\;of\;QD\;emitted\;photons}{\#\;of\;absorbed\;blue\;photons\;by\;the\;QD\;layer}\;=\frac{\#\;of\;photons\;in\;Area\;3}{\#\;of\;absorbed\;blue\;photons\;by\;the\;QD\;layer}$$

Meanwhile, the number of absorbed blue photons by the QD layer can be calculated by comparing Area 1 and Area 2 in the plot. Area 1 represents all the blue photons from the U-shape micro LED, which can be treated as the incident blue photons to the QD layer, and Area 2 represents the blue photons that penetrate (do not become absorbed) through the QD layer. So, the absorbed photons are the difference between Area 1 and Area 2. In addition to the spectral comparison, the reflection happening at the interface of the semiconductor and the air needs to be considered. This reflection could be changed after we deposit a CQD layer due to high refractive index of CQDs. If we take the full structures at the interface, this includes: GaN, 30 nm Al_2_O_3_, and 400 nm SiO_2_ (as explained in the Experimental Section). The additional CQD layer will cause a change of reflection, as shown in Figure 5b. This simulation was performed from the online tool https://www.filmetrics.com (accessed on 21 April 2024). We have the program to calculate two different cases: (a) with CQD and (b) without CQD. As shown in Figure 5b, there is a difference in terms of the wavelength-dependent reflectance for these two structures. Meanwhile, the overall absorption of the CQD layer is proportional to the blue photons that transmit through the semiconductor + dielectric interface. So, we can write down the following equation (assuming no loss during this transmission): *T* = 1 − *R*, where *T* is the transmission and *R* is the reflectance that we calculate in Figure 5b. With the CQD layer, we would expect an increase of the transmitted blue photons due to the better index matching brought by its high refractive index. The ratio of the transmitted blue photons can be expressed as the ratio of the transmission [35]:(8)$$enhancement\;factor=Enh\left(\lambda \right)=\frac{1-R_{QD}\left(\lambda \right)}{1-R_{noQD}\left(\lambda \right)}\;$$

Thus, we can incorporate this *Enh*(*λ*) into the spectral comparison and the final expression becomes [16,36]:(9)$$CCE=\frac{\#\;of\;photons\;in\;Area\;3}{\left(\#\;of\;blue\;photons\;in\;Area\;1\right)-\left(\#\;of\;blue\;photons\;in\;Area\;2\right)}\;=\frac{\int_{QD_{emission}}\left(\frac{\lambda }{hc}\right)\times \left(I_{QD}\left(\lambda \right)-Enh\left(\lambda \right)I_{ref}\left(\lambda \right)\right)d\lambda }{\int_{excitation}\left(\frac{\lambda }{hc}\right)\times \left(Enh\left(\lambda \right)I_{ref}\left(\lambda \right)-I_{QD}\left(\lambda \right)\right)d\lambda }\;=\frac{\int_{QD_{emission}}\left(\frac{\lambda }{hc}\right)\times I_{QD}\left(\lambda \right)d\lambda }{\int_{excitation}\left(\frac{\lambda }{hc}\right)\times \left(Enh(\lambda )I_{ref}\left(\lambda \right)-I_{QD}\left(\lambda \right)\right)d\lambda }$$
where *I_QD_* and *I_ref_* are the spectral responses of the U-shape devices with and without the CQD layer, respectively, and *I_ref_*(*λ*) can be omitted in the numerator because it is almost zero (no QD emission component in the pure blue U-shape LED spectrum). The *QD_emission_* and excitation in the integral limits represent the wavelength range of the CQD emission (>500 nm in this study) and the U-shape LED emission (between 400 to 500 nm). In Figure 5b, the current-dependent CCE can be plotted. Both devices showed their conversion efficiencies higher than 30%: 32.98% for the 60 μm device and 32.95% for the 70 μm device. In the past, when we used a traditional square micro LED with CQD deposited directly on the top of the mesa, the measured results were inferior to this study (8.45% at best) [37], which could be due to the insufficient CQD thickness and thus short optical path for excitation and absorption. In Figure 5c, there is a difference between the low current CCE of the 60 μm device and that of the 70 μm device. In most cases, the CCE proliferates to the maximum at a very low current level and decays gradually as the current heats the device and the CQD layer and curtails the quantum efficiency, as we saw with the 70 μm device current dependence in Figure 5c and the profile (a) in the inset of Figure 5c. However, another potential situation exists when insufficient excitation of the CQD layer happens, which could lead to a different trend of the CCE profile. When the incident blue photons are few and the excitation direction is lateral, the excited CQD emission could be re-absorbed quickly, and this can reduce the CCE. So, if the pumping LED is not strong enough at low current density, we will observe a slow increase of the CCE, as in the 60 μm device case of Figure 5c and the profile (b) of the inset. The CCE will then decrease after the structure receives enough joule heating from the injection currents. Both trends in Figure 5c can be observed in other 60 μm and 70 μm devices, indicating the present uncontrollability of this design. The lateral incidence of blue photons into the CQD layer greatly increases the optical path but also makes the re-absorption and re-emission processes easy. Therefore, the excitation intensity of the U-shape device and the non-radiative processes in the CQD layer become influential and alter the CCE dependence (upward or downward) more unpredictably. This phenomenon is similar to what we previously had for optimizing the QD color conversion layer thickness under constant pumping power [37]. The optimization of this design is essential and shall be our focus in the next phase.

## 4. Results Discussion

In a full-color panel, if we use the color conversion scheme, there will be two different pixels: one is a purely blue color and the other is the color-converted one with a CQD layer. The U-shape micro LED is designed to play the role of the latter. The more blue photons that we block, the purer the overall color of this pixel will be. In a traditional square-shaped micro LED with CQD deposition, we need to make a very thick CQD layer in order to absorb all the blue photons. However, this will hinder both the pixel resolution (or size) and the color conversion efficiency, because it is difficult to pattern the CQD and the CCE will drop if the conversion layer is too thick. So, our mission is to minimize the unabsorbed blue photons in the color-converted pixels and boost the CCE at the same time. In this U-shape device, the top surface is almost covered by the metal contact, which should block the blue emission in the normal direction and redirect the blue photons towards the CQD layer for lateral incidence. The blue photons bounced back by the metal layer and emitted through the inner sidewall of the U-shape structure will have a better chance of being absorbed in the CQD layer at the central courtyard. Thus, we can potentially increase the emission from a thin CQD layer and keep the leaked blue photons low.

### 4.1. SRH Recombination-Related Various U-Shape Micro LEDs

One of the drawbacks of the U-shape micro LEDs is its high ratio of the periphery over the device area (P/S ratio). The situation can often lead to the undesirable high rates of non-radiative recombination which can be seen from the extracted SRH coefficients (A coefficient in Equation (3)) under different device geometries. The effect of a high P/S ratio can be seen in the generic formula for the SRH coefficient [12,34,38]:(10)$$A=A_{0}+v_{s}\times \left(\frac{Periphery}{Surface\;Area}\right)$$
where A_0_ is the non-radiative recombination related to bulk defects and impurities and *v_s_* is the surface recombination velocity. The U-shape devices with different mesa widths (10, 15, and 20 μm) were tested at various current densities, and we used Equation (3) to fit the measured EQE and obtain the SRH coefficients (similar to Figure 3). As we can see in Figure 6, the high SRH coefficients of the U-shape devices lead to a much higher surface recombination velocity and also indicates a higher-than-usual bulk recombination. Compared to its squared devices with ALD coating, whose data was reported in ref. [10], we saw an almost two times (946.06 vs. 551.34 cm/s) increase in v_s_ and 2.8 times (2.37 × 10^6^ vs. 8.44 × 10^5^ s^−1^) increase in A_0_. To reduce this effect, further optimization of the sidewall treatment is necessary. For example, a neutral-beam etch or the KOH chemical etch should be able to improve the performance of these U-shape devices [39,40,41,42].

### 4.2. Low-Temperature EQE for U-Shape Devices

Following the analysis of the previous section, we saw high non-radiative recombination in our U-shape devices. If we can somehow suppress this mechanism, it is possible to see a positive change in the measured EQE. To test this assumption, one of the possible scenarios is to measure its EQE at low temperatures. A 10-micron-mesa U-shape device with a 70 μm × 70 μm courtyard is placed on a cryogenic stage and cooled down by liquid nitrogen. The EQE measured at various temperatures showed a very distinctive feature in Figure 7a. The EQE at 80 K showed a great boost to 3.7% and the J_peak_EQE_ was around 13A/cm^2^. When the stage temperature was raised to 140 K and 200 K, we observed a reduction in the peak EQE (2.42% at 140 K, 2.00% at 200 K) and the increase of J_peak_EQE_ (52.17 A/cm^2^ at 140 K, 152.17 A/cm^2^ at 200 K). By applying the modified ABC model, as we mentioned in Equation (3) [9], we can extract the corresponding SRH coefficients, as shown in Figure 3. The SRH coefficients grow exponentially from 3.57 × 10^6^ s^−1^ at 80 K and 7.95 × 10^7^ s^−1^ at 140 K to 1.20 × 10^8^ s^−1^ at 200 K, and the results are plotted in Figure 7b. In the past, similar results were obtained in a large InGaN device [43], in which the increase of EQE and the shift of J_peak_EQE_ were both expected.

This phenomenon illustrates that a long periphery (or border) of a micro LED can lead to excess non-radiative recombination at elevated temperatures and cause the degradation of the device’s EQE. Thus, the importance of the sidewall passivation can-not be overlooked. The differences between the calculation and measured QE at low current levels (especially at the 200 K profile) can be attributed to the detection limit of our photodetector. At a very low current level, the emitted photons from the U-shape micro LED are few, and the limited coverage of our photodetector could not catch all of them. The other reason could be the extra carrier leakage caused by the parasitic effects [44]. It was reported that the shunt resistance of the LED could significantly increase the inverse of carrier lifetime (non-radiative recombination rate) [44]. If the device has such a carrier path, a lower-than-expected QE can be expected because the constant SRH coefficient (A) cannot handle this situation.

### 4.3. Continuous Aging Tests on the CQD Layer

One of the merits that was brought by the U-shape micro LED is the better thermal distribution when CQDs are deposited in the courtyard area compared to the direct mesa top area of the traditional square devices. As shown in the Experimental Section, we perform the CQD dispense via the AJ inkjet-type of spray and we can drive the micro LED continuously to observe the change of CQD emission. The current density of the micro LEDs can be used as a standard for comparison. In Figure 8a, the CQD emission intensity versus aging time was shown for a directly dispensed CQD layer on a square device at 10 A/cm^2^, a 70 μm U-shape device at 100 A/cm^2^, and a 70 μm U-shape device at 200 A/cm^2^. The QD degradation was evaluated by the peak CQD intensity at the time of the measurement divided by the initial peak CQD intensity (at the 0th minute). So, in Figure 8, the degradation would begin at 100% and decrease as the aging test proceeds. The case with the square device can be named as the traditional or the reference sample. A time duration for the CQD peak intensity dropped to 50% of its initial value will be used to evaluate the effectiveness of the CQD layer. As we expected, the CQD emission in the traditional case dropped to 50% of the initial value in less than 40 min (at the 39th minute in the measurement) into the aging test, while the CQD peak of a U-shape device at 100 A/cm^2^ can last to 1321.26 min (by the linear extrapolation of the experimental data up to the 1080th minute) and 249.11 min at 200 A/cm^2^. The shift of the CQD emission in the traditional case is also large: more than 4.42 nm was recorded during the test, while both U-shape cases only moved slightly (0.033 nm at 100 A/cm^2^, 0.022 nm at 200 A/cm^2^), as shown in Figure 8b. Because the wavelength shift affects the perception of our eyes to the image quality, this illustrates another strength of our design: the color quality of a U-shape CQD device can be maintained even when the CQD is degrading. In other cases, the low-temperature ALD coating or the ionic crystal encapsulation can help protect the CQD layer and increase its life span greatly [36,45]. With proper handling, thousands of hours of CQD lifetime can be reached and this will be crucial for the success of this device in the next phase of development.

## 5. Conclusions

In conclusion, we designed, fabricated, and measured micro LEDs with a U-shape mesa, and we also directly dispensed CQD layers into these U-shape micro LEDs to investigate their photonic properties. The lengthy periphery of the U-shape mesa causes an increase in non-radiative recombination, which can be examined by our EQE measurement and the modified ABC model. An almost two times increase in the U-shape device’s surface recombination velocity can be detected compared to the traditional square micro LEDs. A series of low-temperature (as low as 80 K) EQE measurements further confirm the increase of the SRH coefficient at an elevated temperature. Nonetheless, the best EQE of the U-shape micro LED can reach 6.51% at room temperature. A higher-than-30% CCE was measured after the CQD dispense, and a much longer lifetime of the CQD layer can be obtained due to less direct heating from the LED with this special U-shape mesa. Under the continuous aging condition, the unprotected CQD layer in the U-shape device is expected to last more than 1300 min, but the CQD life span dropped sharply to 39 min in the traditional scheme. While the U-shape device itself needs to be optimized and improved to overcome the stronger non-radiative recombination, the encouraging results from the lifetime improvement of the CQD layer can pave the way for better understanding and integration of the color conversion materials with the semiconductor-based micro LEDs and to construct a highly efficient full-color microdisplay together in the future.

## Figures and Tables

**Figure 1 nanomaterials-14-00938-f001:**
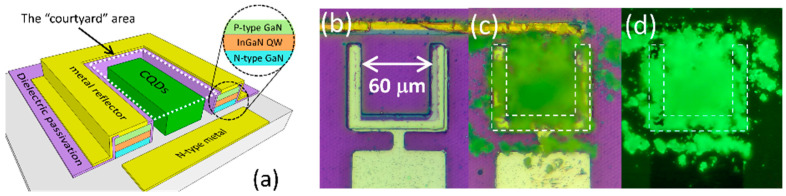
(**a**) The schematic diagram of a U-shape micro LED integrated with a CQD layer in the “courtyard” area (indicated by white dashed lines). (**b**) The U-shape micro LED with a 60 μm by 60 μm courtyard area and a 10 μm wide mesa. (**c**) The same device after the CQD dispense and (**d**) under FLOM. The area circled by white dashed lines indicates the mesa of the U-shape device.

**Figure 2 nanomaterials-14-00938-f002:**
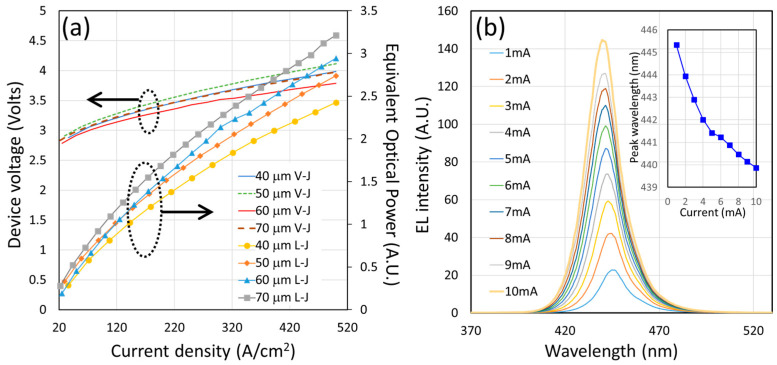
(**a**) The L-J-V measurement of U-shape micro LEDs with different sizes of courtyard area. (**b**) The current-dependent spectra of a U-shape micro LED with a 70 μm by 70 μm courtyard and a 10 μm mesa. The inset shows the current-dependent peak wavelength of this device.

**Figure 3 nanomaterials-14-00938-f003:**
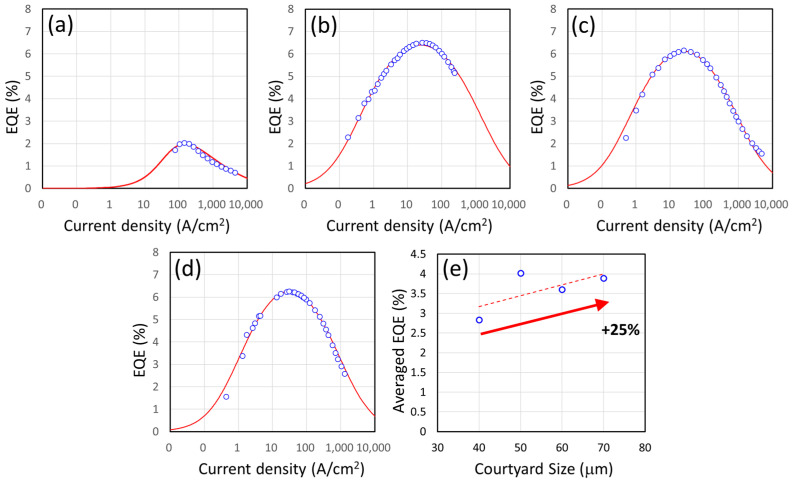
The measured EQE profiles vs. current density of U-shape micro LEDs whose mesa width is between 10 μm and 15 μm and whose courtyards are (**a**) 40 μm by 40 μm, (**b**) 50 μm by 50 μm [20], (**c**) 60 μm by 60 μm [20], and (**d**) 70 μm by 70 μm, respectively. (**e**) The averaged EQE_max_ for various courtyard sizes of U-shape micro LEDs. An increase of 25% can be observed between the 40 μm devices and the 70 μm devices.

**Figure 4 nanomaterials-14-00938-f004:**
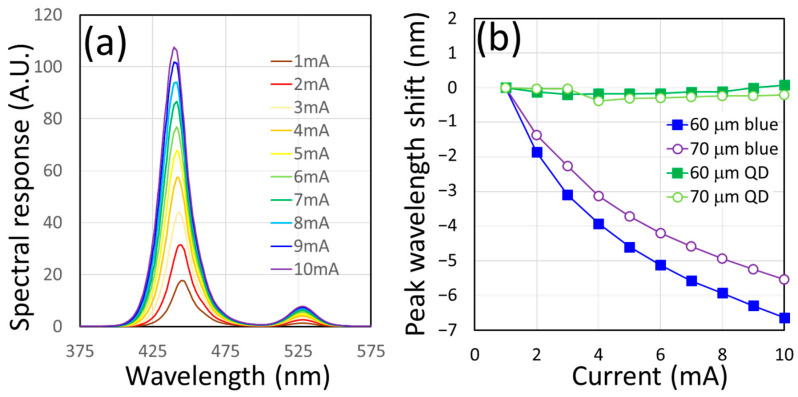
(**a**) The current-dependent emission spectra of a CQD integrated U-shape micro LED with a 10 μm mesa and 70 μm courtyard [20]. (**b**) The emission peaks of the U-shape LEDs and their CQD layers under different currents.

**Figure 5 nanomaterials-14-00938-f005:**
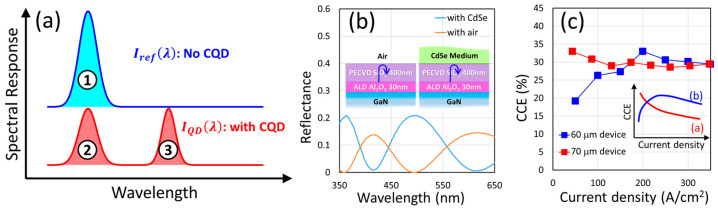
(**a**) An illustration of the CCE evaluation via the measured spectra of the devices with and without CQDs. (**b**) The calculated reflectance of the interface (between U-shape micro LED and air) with and without CQD deposition. This simulation was performed from the online tool https://www.filmetrics.com (accessed on 21 April 2024). The inset pictures depict the two structures: one with QDs (CdSe medium) and the other one with air. (**c**) The CCE under different currents for U-shape micro LEDs with 60 and 70 μm courtyard sizes [20]. The inset illustrates the two common patterns for EQE current-dependency.

**Figure 6 nanomaterials-14-00938-f006:**
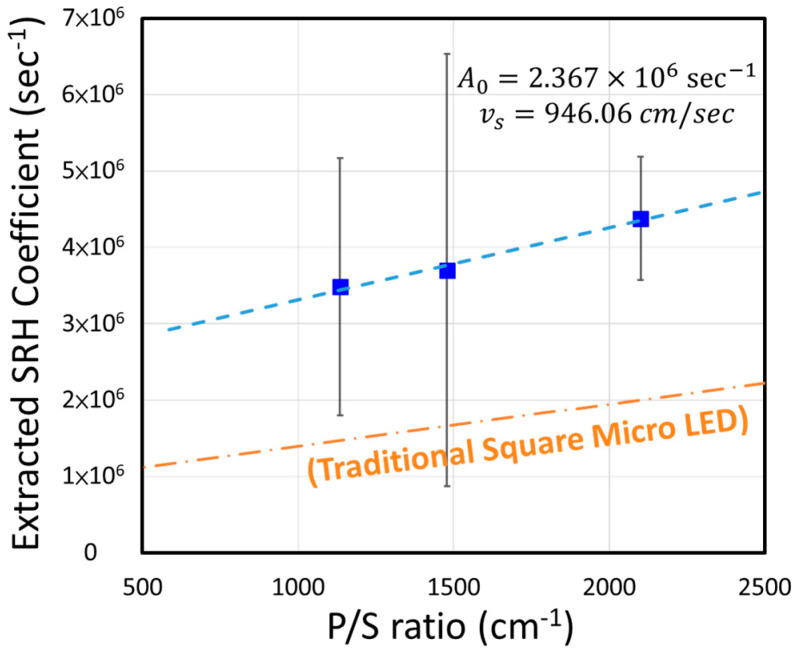
The extracted SRH coefficients versus different periphery-to-surface ratios (P/S ratio). The orange dot–dash line was graphed by the traditional micro LED published in [10].

**Figure 7 nanomaterials-14-00938-f007:**
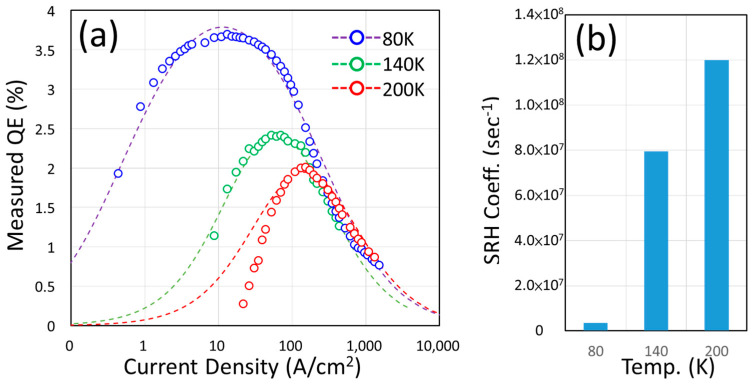
(**a**) The EQE of a U-shape micro LED (with a 10 μm mesa and a 70 μm courtyard) at low temperatures. (**b**) The extracted SRH coefficients in (**a**).

**Figure 8 nanomaterials-14-00938-f008:**
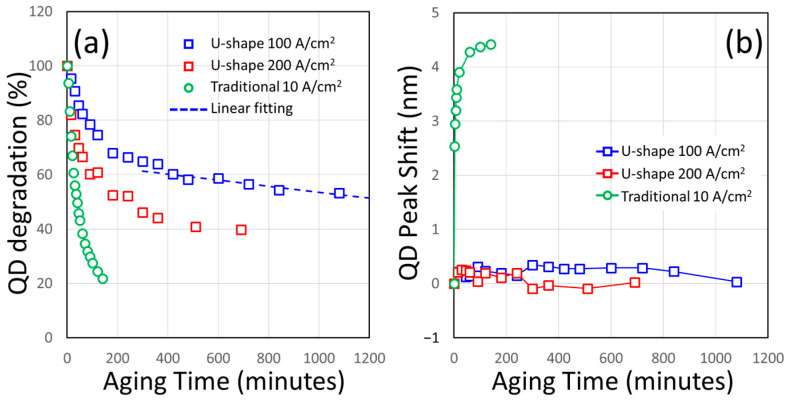
(**a**) The monitored CQD peak intensity during the continuous aging tests. The dashed line is the linear extrapolation of the 100 A/cm^2^ case and it is extended to over the 1300th minute. (**b**) The peak wavelength shifts of CQD emissions during the aging tests.

## Data Availability

The data presented in this study are available on request from the corresponding author.

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
