# Peer review of "A Colloidal-Quantum-Dot Integrated U-Shape Micro-Light-Emitting-Diode and Its Photonic Characteristics"

_nanomaterials, 2024, doi:10.3390/nano14110938_

Round 1

Reviewer 1 Report (Previous Reviewer 3)

Comments and Suggestions for Authors

The paper is devoted to the development of a U-shaped two-spectral micro-LED based on InGaN/GaN QWs, where an array of colloidal QDs is used as a re-emitting layer. This is not my first review of this work. The authors significantly improved the text of the paper and responded to the bulk of comments after the first round of reviewing. However, there are a number of comments about the work. The most significant, in my opinion, is remark No. 10, which suggests that the authors are dealing with an incompletely understood process of creating devices, and uncontrollable factors play a significant role. Nevertheless, I believe that there is scientific and technical novelty in the paper, and after eliminating the comments, the work can be published.

1.    Is anything known about the radiation diagram of the U-LEDs used in the work? What fraction of the QW radiation hits the CQD?
2.    What impurities used for p-doping of GaN? The question is due to the fact that such doping is not very simple (unlike n-type), and requires a separate description.
3.    Do I understand correctly that the InGaN/GaN QW is used only as a pump, and all radiation useful from the point of view of future applications is generated in the QD array? The question arises from the fact that the contact layers on the QW appear to be not optically transparent, so it is doubtful that a significant fraction of the QW radiation will be emitted in the direction of the observer (along the growth axis).
4.    In the title of section 3.1. The abbreviation LIV is not clear.
5.    In Figure 2b and below, the electroluminescence (EL) spectra are shown with the captions «spectral response». In order to avoid inaccuracies in readers’ understanding, I recommend indicating directly the «EL intensity (arb. units)».
6.    Recommendation: in Figure 2b it would be worth making an insert with the dependence of the position of the EL maximum on the current.
7.    Spectra in Fig. 2b measured for the structure without deposited QDs? The question is due to the fact that, as it seems to me, the presence of a QD can affect the electrical characteristics of the device due to the shunting of the QW and the flow of current directly through the QD array.
8.    Are QDs scans susceptible to direct electrical stimulation? Or is luminescence in QDs excited exclusively by QW radiation? This is important for CCE assessment.
9.    I would recommend specifying EQEmax in Figure 3e in the same values as in the other figures in 3. This will eliminate discrepancies when analyzing Figure e, and will not mislead the reader that EQE is close to 100%.
10.    A significant change in the regimes of excitation transfer from the QW to the QD when the mesa size changes from 60 to 70 μm (which, in my opinion, is not a very significant difference) indicates that there are some uncontrollable parameters in the system.
11.    The paper contains 2 Figures 5. Apparently, this is a typo after the first iteration.

Author Response

Reviewer 2 Report (New Reviewer)

Comments and Suggestions for Authors

After reading carefully and thinking about the content of the proposed manuscript entitled “A Colloidal-Quantum-Dot Integrated U-shape Micro-Light- 2 Emitting-Diode and Its Photonic Characteristics” I suggest that this manuscript is suitable for publication in the nanomaterials journal as it is. The authors successfully correct several issues appointed by the referees. The English is sufficient, the scientific sound is excellent, the figures are well-designed and self-explanatory and the novelty of this proposed manuscript is high. The authors have encouraged results demonstrating all the processes of designing and fabrication of micro-LEDS using advanced methodology such as the modified ABC model. What is lacking in the article is some structural characterization but I understand that this paper is more focused on the devices rather than on the fundamental physics.

Author Response

Thank you very much for your encouragement. It is true that this paper is more on the device side than on the physics side. Thanks again.

Reviewer 3 Report (New Reviewer)

Comments and Suggestions for Authors

This paper reports a novel U-shaped GaN blue micro-LED with colloidal quantum dots (CQD) dispensed in the U-area for wavelength conversion. The main conclusion was that the U-shaped design gave rise to a higher color conversion efficiency and a longer lifetime. I feel that the paper is publishable in nanomaterials, but a few revisions are required. 

1)     Fig.3. Current density at which EQE peaks is approximately proportional to nonradiative recombination rate (see, e.g., Ref. 34) and is frequently used as a figure of merit of characterizing nonradiative recombination rates of LEDs. The 40x40 mm2 device showed the lowest peak EQE among the 4 devices given in Fig.3, resulted from a higher nonradiative recombination rate in the 40-mm device as discussed by the authors. On the hand, the 40-mm device showed a current density of about 1 A/cm2 at the peak EQE, much lower than those observed in other devices (~30A/cm2). This is very unusual. I suggest the authors confirming whether they made a mistake in calculating the current density of the 40-mm device. Otherwise, the authors should give a reasonable explanation about the low current density at peak EQE of the 40-mm device.  

2)     Page 7, Eq.8, the calculation of CCE. Light-extraction efficiency of the LED with CQD layer should be higher than that of the device without the CQD layer since the refractive index of the CQD layer should be larger than that of air. The difference in light-extraction efficiency between samples with and without the CQD layer should be taken in account in Eq. 8. 

3)     One of the drawbacks of the proposed approach is that the blue light can not be completely converted into green light and a strong component of blue light remained. However, in display application, one needs a monochromatic micro-LED. Could the authors give some comments on this problem?

4)     Page 3, line 113. Please confirm if 25% is the mistake of 15%.

5)     Page 4, lines 164-166 should be deleted. Please confirm.

Round 2

Reviewer 1 Report (Previous Reviewer 3)

Comments and Suggestions for Authors

The authors take good response on all comments. I think, that the paper can be published now.

This manuscript is a resubmission of an earlier submission. The following is a list of the peer review reports and author responses from that submission.

Round 1

Reviewer 1 Report

Comments and Suggestions for Authors

The paper is dedicated to U-shape micro LED fabrication and integration with colloidal quantum dots. The properties of the fabricated structures were investigated.

More information about the structure preparation should be added. There are no deposition parameters for e-beam, ALD, PECVD etc. Some explanation of parameters in equations are missing (for example 3rd equation).

It would be good that the authors before submission would read the article. Otherwise ther should not be the paragraph:

“This section may be divided by subheadings. It should provide a concise and precise description of the experimental results, their interpretation, as well as the experimental conclusions that can be drawn.”

I would recommend reworking the article with more attention to detail, as well as more precisely defining the purpose of this study and the advantages of the created structure in relation to other similar structures.

Reviewer 2 Report

Comments and Suggestions for Authors

Yu-Ming Jao et al. presented a novel U-shape micro-LEDs are coated 16 with Al2O3 by an atomic layer deposition system, and exhibit EQE of ~6.51% and high surface recombination because of their long peripheries. The findings are interesting, and I recommend considering this work for publication in nanomaterials after addressing the suggestions thoroughly.

1-       I suggest that the authors elaborate further on the introduction section to provide additional details, specifically highlighting the current bottlenecks associated with micro-LEDs.

2-       Does the current method exclusively involve inorganic quantum dots, or could perovskite be a viable alternative? It would be valuable to explore the potential use of perovskites instead of CQDs in this context.

3-       A clarification is needed regarding the synthesis of the CQDs utilized in this study, as the authors have not provided any details on this aspect.

Comments on the Quality of English Language

Minor editing of English language required.

Reviewer 3 Report

Comments and Suggestions for Authors

The paper is devoted to the creation and study of optoelectronic properties of novel U-shaped microLEDs with CQD. It has been shown that this approach to constructing a microLED can increase CCE by almost 4 times compared to traditional square geometry. A significant increase in the service life of microemitters is also shown. The paper leaves a good impression and should be published.
However, there are some comments, mainly related to the style of presentation of the data. Unfortunately, there is some confusion and inconsistency in the data presented. This may be due to the low reproducibility of the data and the significant influence of hidden factors, such as differences in surface recombination rates in specific different structures. In any case, these shortcomings do not reduce the main value of the work.

1.    Line 18. Do the authors mean non-radiative recombination by surface recombination?
2.    It is not clear from the abstract what exactly a U-shaped LED is. Are there other light-emitting areas besides CQD? As will be seen further from the text of the paper, an InGaN/GaN QW is used as a main light emitter. It may be worthwhile to briefly outline this in the abstract in order to give the reader a complete picture of the work.
3.    It is not very clear from the abstract for which cases the LED operating time was compared. For cases of U diode with and without QD? Or for QD diodes with a U diode and with a flat diode (which one)?
4.    As far as I know, to generate light of different wavelengths, it is possible to use low-dimensional structures with absolutely the same composition of layers and differing only in the size of the active region (QW). Shifting the quantum levels due to changing the size of the QW allows one to vary the operating wavelength over a wide range. In this regard, the authors’ statement that to generate different colors requires LEDs of different compositions (line 48-49) seems somewhat incorrect. Moreover, the authors cite compounds that are obviously poorly compatible: InGaN (traditionally having a wurtzite structure) and AlGaInP (sphalerite). This creates the misconception that creating multi-color displays requires solving the daunting task of integrating these compounds.
5.    Intuitively, the U-shape of the LED is designed to improve the geometric emission characteristics and enhance color conversion efficiency. However, it is not clear why the O-form was not used? At first glance, this configuration should be even more efficient. Or it wills the topic of next paper?
6.    What are the dimensions of the InGaN QW in the structure used? It would be appropriate to present these parameters in section 2.
7.    Figure 2a is difficult to understand. I would recommend using arrows to indicate which curves belong to which axes and making the labels more different (in the current version, the line colors are very similar, which makes it difficult to perceive).
8.    Do I understand correctly that the spectra in Figure 2b refer to structures without QDs? Is there any noticeable dependence of the emission spectra of heterostructures on the size of the structures? (I think it shouldn't exist)
9.    It is not entirely clear which solar photocell is discussed in section 3.2 (line 139-140). Was this item used in the measurements? Alas, it is not explicitly described in section 2.
10.    As the authors write on line 116-117, the paper discusses structures with different sizes (40-70 microns). At the same time, experimental data are not presented for all of these sizes. It is advisable to bring these aspects of the paper into conformity.
11.    There is no label for the vertical axis of Figure 4b.
12.    I may be mistaken, but it seems to me that in expression (4) the denominator should only contain the Iref term. Why is Iref-Iqd used there? Please clarify this point.
13.    It is not very clear what causes the strong decrease in CCE at low currents (about 50 A/cm2) for a structure with dimensions of 60 μm, in contrast to a 70 μm structure.
14.    What are the CCE values at lower current values? As previously shown, the optimal current value from the EQE point of view is 25 A/cm2.
15.    The authors discuss EQE at low temperatures (Fig. 6). It can be seen that the value of the maximum EQE (3.5%) is lower than for similar curves in Figure 3 (6%). At what temperature were the measurements presented in Figure 3 taken? How does structure size affect EQE?
16.    What explains the deviation of the experimental EQE data from the calculated curve for 200K (red dots and line in Figure 6 a)?
17.    It is not entirely clear what the scale in Figure 7a means (degradation). Do the authors mean the luminescence intensity of the QDs?
